# Astrocytes Are a Key Target for Neurotropic Viral Infection

**DOI:** 10.3390/cells12182307

**Published:** 2023-09-19

**Authors:** Maja Potokar, Robert Zorec, Jernej Jorgačevski

**Affiliations:** 1Laboratory of Neuroendocrinology–Molecular Cell Physiology, Institute of Pathophysiology, Faculty of Medicine, University of Ljubljana, Zaloška 4, 1000 Ljubljana, Slovenia; 2Celica Biomedical, Tehnološki Park 24, 1000 Ljubljana, Slovenia

**Keywords:** viruses, astrocytes, endosome, lysosome, autophagy, cell metabolism, oxidant species

## Abstract

Astrocytes are increasingly recognized as important viral host cells in the central nervous system. These cells can produce relatively high quantities of new virions. In part, this can be attributed to the characteristics of astrocyte metabolism and its abundant and dynamic cytoskeleton network. Astrocytes are anatomically localized adjacent to interfaces between blood capillaries and brain parenchyma and between blood capillaries and brain ventricles. Moreover, astrocytes exhibit a larger membrane interface with the extracellular space than neurons. These properties, together with the expression of various and numerous viral entry receptors, a relatively high rate of endocytosis, and morphological plasticity of intracellular organelles, render astrocytes important target cells in neurotropic infections. In this review, we describe factors that mediate the high susceptibility of astrocytes to viral infection and replication, including the anatomic localization of astrocytes, morphology, expression of viral entry receptors, and various forms of autophagy.

## 1. Introduction

Infectious diseases of the central nervous system (CNS) are most commonly caused by various types of viruses [1,2]; most of the 25 virus families implicated in human disease include representatives that have been associated with CNS disorders in humans [3,4]. Viruses involved in infection of the CNS can cause inflammation of distinct anatomic regions; inflammation of the meninges is known as meningitis, inflammation of the brain is called encephalitis, and inflammation of the spinal cord is called myelitis; simultaneous inflammation of multiple regions is referred to as either meningoencephalitis or encephalomyelitis [5]. Virus-induced diseases of the CNS may lead to various acute changes in mental and motor functions and cause chronic neurological dysfunctions with long-term consequences that have severe implications for the quality of life of affected individuals (despite an often-mild acute phase) and are associated with substantial mortality. However, viral infections and their long-term consequences are still frequently overlooked, under-reported and under-diagnosed and thus represent a significant burden to human health worldwide [6].

## 2. Virus Spread and Infection of the CNS

The CNS is typically not the site of initial virus entry, except for extrinsic viral contamination of the CNS during head trauma, neurosurgical procedures, via medical implants, and congenital malformations. Primary infections generally occur at more remote anatomic sites. Arboviruses, a group of viruses transmitted to humans by infected arthropod vectors (e.g., ticks and mosquitoes), enter the skin after an infectious insect inoculates the virus into the dermis of a human host (Figure 1A). After cell entry, virus replication may remain localized to the infected cell, or the virus may start spreading beyond the primary site of infection; the infection becomes systemic when it is not contained by the immune system [7]. Factors that restrict an infection from spreading beyond an epithelial surface have been described in detail previously [7]. When spreading occurs, virus progeny produced within the dermis have access to afferent lymphatic vessels through which viruses gain direct access to the permeable lymphatic capillaries of lymph nodes (Figure 1A) or target Langerhans cells [8], which then migrate to and accumulate in the draining lymph nodes [9]. Ingested and inhaled viruses, on the other hand, must first overcome the intrinsic barriers of the mucosal surfaces [10] and establish infection in oropharyngeal or small bowel lymphoid tissues [5]. Once in secondary lymphoid tissues, viruses can enter veins via the efferent lymphatic vessel (Figure 1A) [7]. Circulating viruses are then disseminated via a hematogenous route. During the initial entry of the virus into the bloodstream, which is known as primary viremia, the quantity of circulating virus is typically low, but the ensuing virus replication in target organs may result in the production of higher concentrations of virus in the blood during secondary viremia [7]. In addition to a hematogenous route, viruses can also spread from the primary site of infection to the CNS by retrograde transit within peripheral or olfactory nerve axons after entry into local nerve endings [11,12,13]. Some viruses rely on a specific pathway to enter the CNS (e.g., direct entry of the rabies virus into the CNS using neuronal spread [14]), but many viruses are capable of infecting the CNS in more than one way [15,16].

Viruses circulating in the blood must successfully cross the protective barriers of the brain to establish infection of the CNS. A restricted environment is maintained in the CNS by strict regulation of the entry and exit of substances through the blood–brain barrier (BBB; Figure 1B) and the blood-cerebrospinal fluid barrier (BCSFB), which mediate the immune-privileged status of the CNS [17]. The BBB consists primarily of endothelial cells of microvascular blood vessels, compressed closely to each other by tight junctions (Figure 1B). In contrast, the BCSFB is formed by the network of fenestrated microvessels in the ventricular system and the subarachnoid space [11,18,19].

The BBB was first demonstrated by Paul Ehrlich in the late 1800s when he noticed that acidic dyes injected intravenously stained the tissues of the body but not the brain [19]. Although gases and fat-soluble molecules can diffuse across these barriers, entry of water-soluble molecules, immune cells, and pathogens is limited; even water requires special transporter proteins [19]. Approximately 50 years after the groundbreaking work of Ehrlich, the term “Barrière hématoencéphalique” or BBB was introduced in 1921 by Lina Stern [20] (for an interesting article on the history of BBB permeability studies, see the review by Saunders et al. [17]). Even though the BBB is arguably the most selective and tightly controlled of the CNS barriers, most viruses invade the CNS by this pathway because the BBB is the largest surface area between the brain and the blood, with an estimated combined area of 12–18 m^2^ [7,21,22]. The structural components that form the BBB are endothelial cells, which, unlike other vascular endothelial cells, express tight junction (TJ) proteins that seal the paracellular space between adjoining endothelial cells. These proteins include claudins, occludin, tricelllin, lipolysis-stimulated lipoprotein receptor, junctional adhesion molecules, and zonula occludens proteins; the latter form a structural link to the actin cytoskeleton and actin-binding proteins. In addition to endothelial cells, astrocytes and pericytes are critical for the development and maintenance of the BBB by regulating the TJs [23,24,25] (see also Figure 1B). To cross the BBB, viruses have developed diverse strategies that can be classified into three main categories: transcellular, paracellular and/or the “Trojan horse” mechanism(s) (Figure 1B) [26]. Transcellular traversal involves infection of endothelial cells, in which viruses may or may not replicate, followed by direct passage into the CNS [27,28]. In the paracellular crossing of the BBB, viruses migrate between the proteins comprising the TJs (occludin and claudin proteins that use scaffolding and regulatory proteins to link to the cytoskeleton) [11,26]. Systemic viral infection can lead to increased cytotoxic effects and induce inflammation, mediated by various cytokines, chemokines, free radicals and matrix metalloproteinases, which results in disruption of the TJs and the consequent breakdown of the BBB, allowing viruses unimpeded access to the CNS [11,29,30]. The Trojan horse mechanism of viral infection of the CNS refers to the haematopoietic cells (e.g., leukocytes; Figure 1) infected with a virus that transports the virus into the CNS [26,28,31,32]. After crossing the BBB, viruses may disperse through the CNS [33].

## 3. Astrocytes as the Key Cell Type of CNS Viral Infections

Once a virus enters the CNS, it is considered neuroinvasive. In contrast, neurotropic viruses can infect and successfully replicate in cells of the CNS, and neurovirulence indicates the ability of a viral infection to cause CNS pathology [34,35]. A high degree of cellular heterogeneity of the CNS enables neuroinvasive viruses to infect a variety of cell types, including but not limited to different subtypes of neurons, astrocytes, oligodendrocytes, microglia, pericytes, and ependymal cells. Due to their abundance, role in communicating information and their inherent poor ability to regenerate, most studies investigating neuroinvasive viruses have focused on mature neurons of the CNS. In the last decade, interest in other cell types, especially astrocytes, has grown considerably ([19,21,36,37,38,39,40] and references within), but the pathophysiology of viral infection in astrocytes remains poorly understood. However, several lines of evidence support the importance of astrocytes in viral infections of the CNS: astrocytes (i) are one of the first cell types to be infected after viruses cross the BBB; (ii) are almost as numerous as neurons [41] and have a surface area-to-volume ratio that is fourfold higher than that of neurons [42]; (iii) express several viral receptors (as discussed in detail later); (iv) are highly metabolically active and are closely coupled metabolically with neurons [35,43,44,45]; and (v) can produce and release high amounts of viral progeny [38,40,46]. The relatively high level of virus production in astrocytes likely relies on the properties of astroglial aerobic glycolysis [47]. Here, the end product of aerobic glycolysis is L-lactate, despite adequate oxygen levels, a phenomenon typically present in cancer cells known as “the Warburg effect” [48].

As discussed, astrocytes are one of the first cell types that a virus encounters when crossing the BBB. The basic morphological backbone of astrocytes is similar to that of neurons. Yet, detailed morphology reveals cell expansions that may branch to thousands of smaller processes [42], enabling astrocytes to have a much larger membrane interface with the extracellular space. Extracellular space, estimated to occupy ~20% of brain tissue [49], during infections contains complete viral particles and extracellular vesicles containing infective viral genomes and quasi-enveloped viruses [45]. In addition, astrocytes have been shown to express numerous viral entry receptors important for the preferential targeting certain viruses to particular cell types.

## 4. Viral Attachment to Entry Receptors in Astrocytes

Viral entry into astrocytes, like other cell types, is mediated by the attachment of viral surface proteins to receptors at the plasma membrane. These receptors are involved in various cellular physiological processes; however, viruses exploit them to enter their host cell. Potential viral entry receptors in astrocytes are composed of diverse proteins (Figure 2); a given virus may use several different receptors, as evident from Table 1. Concomitantly, individual viruses can exploit several receptors for entry. Most potential entry receptors summarized in Figure 2 have already been confirmed to mediate astrocyte entry (Table 1).

### 4.1. Confirmed Viral Entry Receptors in Astrocytes

#### 4.1.1. Heparan Sulphate Proteoglycans

Heparan sulphate proteoglycans (HSPGs) are a prominent group of receptors and have been confirmed to mediate viral entry into astrocytes. These receptors comprise a core protein covalently linked to glycosaminoglycan chains formed by unbranched sulphated anionic polysaccharide heparan sulphates [56]. HSPGs comprise syndecans (the core protein is composed of an extracellular domain, a single transmembrane domain, and a short cytoplasmic domain that interacts with the cell cytoskeleton) and glypicans (the core protein is anchored in the cell membrane via glycosylphosphatidylinositol [GPI]) [56,57]. HSPGs are present at the plasma membrane of almost all eukaryotic cells and are involved in many cellular processes, e.g., they mediate interactions with extracellular factors regulating cellular adhesion, play a role in endocytosis, lysosomal degradation, transcellular transport, regulation of cell motility and growth, and affect intercellular signalling and development [56,57,58,59,60]. On the one hand, stress can lead to proteolytic cleavage and shedding of HSPGs.

Conversely, conditions mimicking injury response promote higher sulphation and upregulation of their expression [57,61]. The latter has been shown after viral infection [56]. In astrocytes, the involvement of HSPGs in the entry of viruses has been corroborated extensively. In the case of herpes simplex virus type-1 (HSV-1) infection of astrocytes, the interaction between HSV-1 and HSPGs leads to Ca^2+^-dependent release of ATP. Released ATP activates purinergic P2 receptors (P2YR) at the plasma membrane of astrocytes and neighbouring neurons alike, evoking Ca^2+^-dependent activation of glycogen synthase kinase (GSK)-3 [50]. GSK-3 is a versatile enzyme involved in many functions, including glycogen metabolism, insulin signalling, inflammatory response, and innate immunity [62]. GSK-3 was shown to be essential for phosphorylation of the severe acute respiratory syndrome coronavirus (SARS-CoV) SARS-CoV-2 nucleocapsid protein, and GSK-3 inhibition blocked SARS-CoV-2 infection in human lung epithelial cells [63]. A similar GSK-3-dependent molecular pathway underlying HSV-1 infection is likely shared by neurons and astrocytes, given that the blockade of GSK-3 activation inhibited infection of both cell types [50]. Of note, in human neuronal cells, HSV-1 upregulates GSK-3, which hyperphosphorylates tau protein [64]. In addition to being implicated in the accumulation of amyloid beta peptide (Aβ), dysregulation of calcium homeostasis, and impaired autophagy, this is one of the mechanisms by which HSV-1 could act as a causative agent in the pathogenesis of sporadic Alzheimer’s disease.

The finding that HSPGs are expressed more abundantly in astrocytes in comparison with neurons [50] is consistent with a previous proposal that astrocytes are among the first cells to get infected by viruses that enter the CNS and that astrocytes are a major producer of certain viruses, as was confirmed for Zika virus (ZIKV) [38]. An increased presence of HSPGs at the plasma membrane of HSV-1-infected astrocytes [50] is in line with the prediction that HSPGs are also involved in attachment and entry of tick-borne encephalitis virus (TBEV) into astrocytes; increased time-dependent entry of pre-labelled TBEV and ZIKV was demonstrated, which suggests a positive feedback loop for viral entry [38,40].

Although HSPGs have an important role as a primary attachment molecule that concentrates viral particles on the cell surface and may facilitate subsequent binding to more specific receptor molecules, HSPGs may also mediate viral entry directly [60]. Nevertheless, HSPGs have been confirmed to serve as direct entry receptors only in rare cases, as described for HSV-1 [65].

#### 4.1.2. Angiotensin Converting Enzyme 2

In recent years, angiotensin converting enzyme 2 (ACE2) has gained significant attention because it enables the binding and entry of severe acute respiratory syndrome coronaviruses SARS-CoV and SARS-CoV-2. ACE2 is a type I transmembrane protein composed of an extracellular heavily N-glycosylated N-terminal domain containing the carboxypeptidase site and a short intracellular C-terminal cytoplasmic tail (Figure 2) [66]. ACE2 is an ectoenzyme located on the surface of endothelial and other cells. Although its primary binding partner and substrate appears to be the hormone angiotensin II, ACE2 can hydrolyse other physiological substrates, including inactivation of bradykinin receptor, as well as modulate various physiological processes, such as regulation of renal amino acid transport, pancreatic insulin secretion, control of the transport of intestinal neutral amino acid transporters of the B0AT1 family, mediation of intestinal inflammation and diarrhoea. In the CNS, it mediates cleavage of Aβ, and acts as the receptor for SARS viruses [67,68]. In the brain, ACE2 is expressed predominantly in the cerebral microvasculature, in the medulla oblongata, the hypothalamus, the subventricular zones, and the meninges around the medulla oblongata and hypothalamus [36]. At the cellular level, ACE2 is expressed primarily in astrocytes, rendering them highly susceptible to SARS-CoV-2 infection [36,69]. The S protein of SARS-CoV-2, essentially a crown-like (corona) trimer, facilitates the binding of the virus to ACE2 as well as membrane fusion [70]. Formation of a fully assembled infectious SARS-CoV-2 virion in its prior host or the extracellular space requires proteolytic cleavage of the S protein by furin or furin-like proprotein convertase into the S1 and S2 subunits. This cleavage occurs during biosynthesis and maturation in the Golgi apparatus of infected cells. Following cleavage, the S1 and S2 subunits of SARS-CoV-2 remain non-covalently associated. During the infection of a target cell, the receptor binding domain within the N-terminal end of the S1 subunit binds ACE2, and the S2 subunit anchors the S protein to the membrane and mediates membrane fusion [70,71,72]. Specifically, binding a SARS-CoV-2 virion to ACE2 exposes a site of the S2 subunit termed the S2′ site. This site of the S2 subunit is then further cleaved (acid-dependent proteolytic cleavage) by type II transmembrane serine protease (TMPRSS2) at the cell surface [73] or by cathepsin L in the endolysosomal compartments [74]. As a result, a fusion peptide is exposed and inserted into the membrane. In this process, an antiparallel six-helix bundle is formed, which allows the mixing of viral and cellular membranes, resulting in the fusion and consequent release of the viral genome into the cytoplasm [52,75,76]. Even though ACE2 expression has been confirmed in astrocytes on several occasions and is believed to be the prime receptor for SARS-CoV-2 [36,77,78], other proteins mediate SARS-CoV-2 entry.

#### 4.1.3. Neuropilin-1

Neuropilins (NRPs) are cell surface transmembrane glycoprotein co-receptors for class 3 semaphorins, polypeptides with key roles in axonal guidance, and members of the vascular endothelial growth factor family of angiogenic cytokines [79]. The primary physiological role of NRP-1 is essential for neuronal and cardiovascular development; in the case of disease, it initiates signal transduction pathways involved in cell proliferation, migration, survival, and cancer invasiveness (of various cancer cells, including melanoma, breast cancer and glioblastoma [80,81]).

Moreover, NRP-1 was recently identified as a receptor for SARS-CoV-2 [82], with confirmed high expression levels in astrocytes and neurons [51]. The current understanding of NRP-1 mechanisms in viral infections (especially with SARS-CoV-2) includes (1) recognition of S protein, (2) facilitation of the membrane fusion, (3) possible modulation of endocytosis, and (4) modulation of immune responses. NRP-1 has been identified as a co-receptor for certain enveloped viruses, including the SARS-CoV-2. Namely, the S protein of the virus interacts with NRP-1 in addition to the primary receptor ACE2 [51]. NRP-1 not only promotes viral entry through S protein binding but was also suggested to aid in the fusion of the viral membrane with the host cell membrane, possibly even in the absence of ACE2 [83,84]. Following viral entry, it is likely that NRP-1 plays a role in directing the viral cargo to specific cellular compartments [85]. NRP-1 is also implicated in the immune response to viral infection, as NRP-1 is expressed on regulatory T-lymphocytes, and the therapeutic manipulation of these cells depends on the surface receptors, including NRP-1 [86,87].

Like ACE2, NRP1 is known to bind furin-cleaved substrates and has recently been demonstrated to function as one of the primary viral entry receptors in astrocytes in the case of SARS-CoV-2 [51]. While no studies addressed detailed mechanisms of NRP-1 mediated SARS-CoV-2 entry into astrocytes, binding of S protein to NRP-1 is enabled by aforementioned furin cleavage [51]. Namely, cleavage of S generates a polybasic Arg-Arg-Ala-Arg carboxyl-terminal sequence on S1, which conforms to a C-end rule motif that binds to cell surface NRP1 and enhances SARS-CoV-2 infection, as demonstrated in Caco-2 cells, a human colon adenocarcinoma cell line endogenously expressing ACE2 [88]. As Kong et al. [51] pointed out, measuring the absorption rate and dissociation constants for the interaction of different spike proteins with ACE2 versus NRP1 will better define the molecular basis of apparent differences in receptor utilization. Further research of NRP-1-induced modulations of specific signal transduction pathways in viral infection is warranted, as these pathways (e.g., the Akt and the MAPK/ERK signalling pathways) are associated with cell proliferation, survival, and migration [80,81].

#### 4.1.4. TIM and TAM Phosphatidylserine Receptors

Various RNA and DNA-enveloped viruses exploit phosphatidylserine (PtdSer) receptors to enhance their entry into host cells. In the case of enveloped viruses, the nucleic acid and the capsid are enclosed within the envelope, derived from virus-encoded membrane-associated proteins and partly from modified host cell membranes acquired by the virus during its egress from the host cell [89]. The plasma membrane and a wide variety of intracellular membranes can serve as platforms for virus budding and egress, e.g., the nuclear envelope, rough and smooth endoplasmic reticulum (ER), endosomes, and Golgi compartment [89]. Hence, PtdSer in these membranes ends up in the viral envelope, enhancing virus binding and entry into host cells via PtdSer-mediated virus entry-enhancing receptors (PVEERs) [90,91]. PVEERs are broadly expressed across various tissues, including the CNS, and comprise TIM family proteins, TAM family receptors (AXL, TYRO3, Mertk), and MFG-E8/integrin αvβ3 or αvβ5 complexes [90].

All these receptors have been identified in astrocytes [30,92], but not all have been linked to viral entry. Among these, the receptor tyrosine kinase AXL and its ligands, growth-arrest-specific six and protein S, are implicated in multiple cellular responses, including cell survival, proliferation, migration, and adhesion [93,94]. In astrocytes, AXL has been identified as the primary entry receptor for ZIKV [92]. Compared with neural progenitor cells and neurons, where AXL expression is largely absent during developmental stages in humans, AXL becomes abundantly expressed in astrocytes and radial glia in the mid-gestation period, and it may play an important role in the replication and spread of ZIKV in the fetal CNS [95,96,97]. Astrocytes internalize significantly higher amounts of ZIKV, replicate more ZIKV and tolerate higher viral loads than neurons [38]. AXL has been proposed to promote ZIKV infection in astrocytes by antagonizing the type I interferon signalling [53]. In astrocytes, β-catenin, the central mediator of canonical Wnt signalling, negatively regulates the transcription of AXL but not TYRO3, thus limiting ZIKV internalization [92]. However, ZIKV bypasses this line of defence by downregulating β-catenin to facilitate AXL expression [98]. In comparison with AXL, other PtdSer receptors in astrocytes, such as TYRO3, TIM-1 and dendritic cell (DC)-specific intercellular adhesion molecule-3-grabbing nonintegrin (DC-SIGN) (Figure 2), have limited potential in mediating ZIKV entry due to their minuscule expression [92].

#### 4.1.5. Integrins

Integrins, which are cell surface extracellular matrix receptors, can also be exploited by viruses and bacteria to invade cells [54]. Astrocytes express several integrins, e.g., αvβ3 [99], αvβ5 and αvβ8 [100], which may participate in mediating viral entry in these cells. Viruses from the Picornaviridae and Herpesviridae families, some of which were shown to infect astrocytes, interact with several members of the integrin family, including αvβ3, αvβ6, α2β1, α5β1 in the case of picornaviruses and β1, αvβ3, α3β1, α2β1 in the case of herpesviruses [54]. These interactions may affect downstream signalling events and activate an innate immune response; e.g., an interaction between αvβ3 with toll-like receptor 2 has been demonstrated in a 293T human kidney cell line infected with HSV-1 [101]. Integrin receptors are also exploited by viruses that infect the respiratory epithelium because they facilitate virus entry into cells and modify the induction of intracellular signalling, which has a profound effect on disease pathogenesis [54]. The participation of integrins as potential viral entry receptors in astrocytes has not yet been addressed.

#### 4.1.6. Cluster of Differentiation 147 and Dipeptidylpeptidase 4

Transmembrane glycoproteins from the immunoglobulin superfamily cluster of differentiation 147 (CD147) and dipeptidylpeptidase 4 (DPP4, also known as CD26) are expressed in the human cortex and have been demonstrated as MERS-CoV and SARS-CoV-2 receptors [102,103,104]. The spike RBD of coronaviruses binds with high affinity to CD147 and DPP4 [104,105]. Expression of DPP4 and CD147 is known to increase in response to inflammation in reactive astrocytes. Both proteins appear to be upregulated in infected astrocytes [102,106,107], and as such, they might significantly contribute to viral attachment and entry into astrocytes. Therefore, the upregulation of DPP4 and CD147 may serve as a positive feedback mechanism during viral infection.

## 5. Viral Entry and Remodelling of Intracellular Organelle Traffic in Infected Astrocytes

Viral infection of astrocytes affects several cellular processes, altering their morphologic and physiologic properties, including rearrangement and reorganization of the cytoskeleton, modulation of vesicular traffic with endocytosis and cellular organelles, affecting the transcription of genes with an altered expression profile of cytokines, affecting autophagy and resulting in damage to cellular DNA [40]. Numerous neurotropic viruses may infect astrocytes and indirectly alter the cellular functions of neighbouring neurons and the BBB-forming endothelial cells ([108] and references within).

The viral replication cycle is comprised of four major steps: (1) attachment and entry into a target cell, (2) replication of the viral genome, (3) synthesis of viral proteins and genome packaging into infectious progeny, and (4) egress to the extracellular space and dissemination to the next target cell. All of these steps are affected by the dynamics of the transport and remodelling of intracellular membrane-bound compartments.

In general, viruses hijack the cytoskeletal macromolecular transport networks of the host cell to facilitate their movement across the cytoplasm [109]. Here, another positive feedback mechanism is present, in which, following virus entry, the remodelling of actin filaments and microtubules secondarily affects endocytosis and the ensuing viral invasion into cells. Several neurotropic viruses prominently exploit endocytosis in astrocytes as their entry mode [38,40]. In the case of certain flaviviruses (e.g., TBEV, ZIKV), an increase in the number of intracellular viral particles in a time-dependent manner is accompanied by the increase in speed and directionality of the endocytosed virus-laden vesicles [38,40]. In the case of endocytotic entry of ZIKV, binding to the AXL/Gas complex represents an important, although not exclusive, pathway of internalization [95]. Increased production of new infective virions in cells with increased internalization of endocytosed ZIKV and TBEV [38,40] implies that viruses mainly escape the degradation pathway, as the viral RNA is released from endocytic vesicles into the cytosol before the fusion with lysosomes. In contrast, after entry into astrocytes via the endocytotic pathway, HIV-1 gets predominantly degraded in lysosomes, and only a few virions are released to spread the infection [110].

In the case of infection of astrocytes with SARS-CoV-2, the two-pore cation channel 2 (TPC2) protein also plays a prominent role in the spread of the virus [51]. TPC2 is a member of the voltage-gated ion channel superfamily encoded by the *TPCN2* gene, which regulates the physiological functions of the endolysosomes [111]. TCP2 is localized at the endolysosomal membrane and controls the ionic homeostasis of these acidic organelles [112]. Physiologically, TPCs regulate several processes, including coupling endolysosomal function with the metabolic state of a cell and with autophagy [112]. During infection, TPC2 appears responsible for controlling the movement of viral particles through the endolysosomal system, thus relocating the virus from the endolysosomal network into the cell cytoplasm [51,113,114]. Namely, TPC2 mediates Ca^2+^ release from endosomes/lysosomes activated by nicotinic acid adenine dinucleotide phosphate. On the one hand, released Ca^2+^ regulates membrane fusion events through Ca^2+^ sensors (i.e., synaptotagmins) involving the soluble N-ethylmaleimide-sensitive fusion protein attachment protein receptors. On the other hand, it controls the direction and specificity of trafficking processes via Rab effectors and (Ca^2+^ sensors) synaptotagmins [115,116].

The pattern recognition receptor DC-SIGN also participates in viral endocytosis. DC-SIGN recognizes carbohydrate structures on viral glycoproteins and functions as an attachment factor for several enveloped viruses [117]. To date, only HIV-1 has been shown to be endocytosed in astrocytes via the DC-SIGN receptor, predominantly resulting in virus degradation [55]. In contrast, in dendritic cells, upon internalization and processing of HIV-1 via DC-SIGN, viral exogenous antigens are cross-presented on MHC class I, which induces anti-HIV cytotoxic T-cell responses. It is unclear how various viruses are engaged in DC-SIGN-mediated endocytosis in astrocytes.

## 6. Autophagy in Viral Infection of Astrocytes

Once in the cytoplasm, viruses commence their replication cycle by interacting with cellular membrane compartments to induce cytoplasmic membrane structures, in which genome replication and assembly occur [118]. Plus-stranded RNA viruses and large DNA viruses, in particular, cause large membrane structures originating from the ER or endosomes that support the replication of their genomes [118]. The ER is also associated with early autophagic structures [119,120] that have been proposed to promote viral replication in certain cases. The physiological significance of autophagy in astrocytes concerning the life cycle and transmission of viruses remains poorly investigated. Several flaviviruses (e.g., dengue virus (DENV), West Nile virus (WNV), chikungunya virus CHIKV and ZIKV) were assessed for their possible involvement in autophagy in various cell types, although rarely in astrocytes and neurons. Experimental evidence suggests that autophagy may act anti-viral or pro-viral, depending on the strain of the virus and the type of infected cell [121]. Judging by the limited number of studies conducted on astrocytes, autophagy may also have a dual role in this cell type.

Regarding the anti-viral role of autophagy, virus-induced autophagy helps host cells counteract arbovirus infection and replication by degrading the virus and enhancing anti-viral immunity by augmenting antigen presentation [121,122]. In virus-infected host cells, autophagy also initiates an innate immune response and production of antiviral cytokines. Viral pathogen-associated molecular patterns can be detected by host cell pattern recognition receptors, which initiate anti-viral responses by recruiting signalling molecules that promote the generation of type I interferon (IFN-I) and other pro-inflammatory cytokines [123]. IFN-I is a crucial anti-viral factor in astrocytes; it restricts viral growth via autocrine and paracrine signalling through the interferon-α/β receptors, as demonstrated after infection of astrocytes with TBEV, JEV, WNV, and ZIKV [124]. When autophagy was monitored in human astrocytes, induction of autophagy was detected after infection with TBEV and WNV (Figure 3) [46]. However, inhibition of the formation of autophagic structures does not affect viral replication or the production of infective viruses [46]. Thus, in these cells, remodelling of the ER with the unfolded protein response pathway, not autophagic structures, appears to be a decisive factor for flavivirus replication [46]. Viruses have evolved to exploit evolutionary conserved stress responses, such as the ER unfolded protein response (UPR^ER^), to manipulate cell stress and metabolic pathways, thereby enhancing infection and progeny formation or inducing cell death [125]. Several viruses have been shown to induce the UPR^ER^ by causing ER overload or by eliciting specific signals originating from the ER, as recently reviewed in [125].

In contrast to the induction of autophagy in astrocytes by TBEV and WNV, HIV-1 infection attenuates autophagy in astrocytes via the Nef protein, which blocks the last step of autophagy and thereby helps HIV-1 virus avoid autophagic degradation [126,127]. Modulation of autophagy in astrocytes by some viruses (e.g., TBEV and WNV), but not all (MOF), is evident from Figure 3.

### 6.1. Autophagy of Cellular Organelles in Viral Infection

In most cases, viral infection induces severe damage to various intracellular organelles, thus initiating selective autophagy to degrade dysfunctional organelles to maintain cell homeostasis. Depending on the organelle targeted for the elimination (e.g., mitochondria, peroxisomes, ER, lysosomes, lipid droplets), selective autophagy comprises mitophagy, pexophagy, reticulophagy, lysophagy, and lipophagy [128]. Selective autophagy can be utilized by a virus to promote its replication or help the host cell combat infection. Yet, the role of autophagy in the formation of virus-modified membranes that are involved in replication remains debated [118,128]. For example, mitophagy, which controls mitochondrial quality by inducing clearance of damaged mitochondria via the autophagic machinery, can be promoted by diverse viruses to favour their replication in host cells [129]. Various strategies have been revealed for how viruses exploit mitophagy to facilitate replication. For instance, degradation of autophagy-mediated mitochondrial antiviral-signalling protein (MAVS) has a pro-viral effect because MAVS-dependent activation of nuclear factor-kappa B (NF-κB) and IFN-regulatory factor (IRF3) is hampered and consequently type I IFN anti-viral action is impaired [108,129]. Also, mitophagy was proposed to, on the one hand, promote cell survival by inhibiting oxidative stress and apoptosis and, on the other hand, facilitate persistent viral infection [130,131]. So far, only one study has addressed the role of mitophagy in viral infection of astrocytes. During HIV-1 productive infection of astrocytes, mitophagy was shown to be crucial for cell death resistance and avoidance of inflammasome activation [132]. This study suggested that the inhibition of mitophagy in astrocytes may represent a potential therapeutic strategy aimed at reducing or eliminating HIV-1 reservoirs in the brain [132].

Although canonical macroautophagy is supposed to act pro-virally by viruses high jacking autophagic structures as viral replication sites [133], it also serves as a potent anti-viral pathway [120]. When a virus exploits the ER for its replication, newly synthesized viral proteins aggregate in the ER. Excessive protein processing in the ER inevitably culminates in reticulophagy, inhibiting viral protein synthesis and thus attenuating viral replication. As demonstrated in human brain microvascular endothelial cells, reticulophagy has been reported to assist resistance to infection and limit the replication of Ebola, dengue virus and ZIKV [134]. Yet, to promote their replication, viruses evolved to evade and subvert reticulophagy by inhibiting the ER-resided FAM134B receptor. This receptor binds to membrane-bound LC3 and promotes the docking of ER-derived vesicles to autophagosomes. Its inhibition thus impairs the reticulophagy-dependent clearance of viruses [120,135]. Relationships between reticulophagy and viral infections are poorly explored and thus far completely unexplored in astrocytes. Given that the ER is one of the major organelles that governs the replication and assembly of certain viruses (e.g., polyomaviruses) by penetrating the ER membrane to gain access to the cytosol and then the nucleus [136] and that astrocytes are major viral producers in the CNS, modifications and dynamics of ER inflicted by viral infection in astrocytes deserve attention in the future.

### 6.2. Biogenesis of Lipid Droplets and Lipophagy in Viral Infection

Lipid droplets (LDs) are dynamic intracellular lipid storage organelles that respond to the physiological state of cells. They participate in the regulation of cell metabolism and play a protective role in many cellular stressors, including oxidative stress [137,138]. In a pathological state of the CNS, the content of LDs increases, particularly in astrocytes and microglia, where LDs are speculated to support energy provision and to act neuroprotectively [137,139]. LDs could have both beneficial and detrimental effects in the context of neuroprotection, yet the exact mechanisms and overall impact are not yet fully understood. A potential neuroprotective action of glial LDs was proposed as a failure to form them resulted in more severe neurodegeneration [140,141]. For example, during hypoxia, the overproduction of saturated fatty acids generates ceramides and acylcarnitines that lead to lipid-mediated toxicity and contribute to cellular dysfunction and neuronal death [137]. The formation of LDs can buffer saturated fatty acid build-up in the cell, thereby reducing toxicity [137]. Understanding the roles of LDs biogenesis and lipophagy in viral infection is also elusive. While some studies report that LD build-up in virus-infected cells may reduce viral toxicity and that LD build-up can enhance viral replication, others report that LD build-up can reduce viral replication by activating antiviral innate pathways. For example, a study in baby hamster kidney fibroblasts (BHK cells) demonstrated that infection with DENV increases the number of LDs, some of which are enriched with dengue virus (DENV) viral capsid protein [142]. Thus, LDs are speculated to provide a platform for the formation of viral nucleocapsid and enhance viral replication, a speculation supported by the finding that pharmacological inhibition of LD formation reduced DENV replication [142].

In contrast to assisting viral replication, LDs were also suggested to play a pivotal role in protecting the host against viral infection [143]. A study by Monson et al. [144] in astrocytes showed that enhanced biogenesis and accumulation of LDs in HSV-1 and ZIKV-infected primary human foetal immortalized astrocytes corresponds with enhanced cellular type-I and -III IFN production, thereby decreasing viral replication. Contrary to LD biogenesis, lipophagy, the autophagic degradation of lipid droplets and an essential mechanism for maintaining homeostasis of LDs, can also be directly activated by viruses [145]. Lipophagy involves protein machinery common to macroautophagy, such as microtubule-associated proteins 1A/1B light chain 3B (LC3) [137,142]. As demonstrated in a human hepatoma cell line (HepG2 cells), DENV infection induces lipophagy via activation of 5′-AMP-activated protein kinase catalytic subunit α2 (AMPK), which in turn inhibits the mechanistic target of rapamycin 1 (mTORC1) [122]. These steps result in a depletion of stored triglycerides and an increase in β-oxidation and energy production [146,147]. An increase in ATP boosts the synthesis of the biomolecules needed for viral replication. ATP is required for the receptor-mediated attachment of viral particles and their entry into the cytoplasm [148]. Thus, the role of LDs in the viral cycle of astrocytes remains to be fully elucidated.

## 7. Conclusions

Newly emerging neurotropic viruses and the expanding geographic area of existing viruses represent a huge health burden and a major worldwide concern. It is becoming increasingly evident that to address the severe neurological conditions that can be elicited by neurotropic viruses, further research that significantly advances the boundaries of current knowledge is needed. With this in mind, attention is turning more and more to astrocytes. These glial cells are important as a viral reservoir, producers of new infective virions, and modulators of the immune response in the CNS. Despite the immense importance of astrocytes in neurotropic viral infections, only a handful of studies have systematically addressed the impact and exploitation of cellular processes by viruses in this cell type. Future research should focus on the possible targeting of entry receptors and specific cellular processes to limit virus-triggered malfunctions of astrocytes and the CNS. Of note, understanding the mechanisms of virus entry and replication in astrocytes may lead to therapeutic strategies that would reduce their capacity as a CNS viral reservoir and help prevent virus-associated astrocyte demise. This would be a major step forward, as currently, no effective antiviral drugs are available for many viral infections, nor are we in possession of astrocyte-specific therapies or drugs.

## Figures and Tables

**Figure 1 cells-12-02307-f001:**
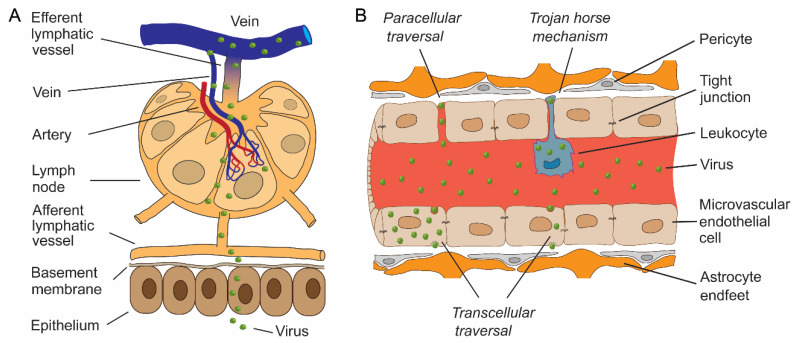
Entry of viruses through the epithelial barrier, the lymphatic spread of viruses and neuroinvasion through the blood–brain barrier. (**A**) After entry through the epithelium, viruses may continue with subepithelial invasion by exploiting the network of lymphatics to enter lymph nodes. Viruses enter porous lymphatic capillaries and/or efferent lymphatic vessels to access the venous system. Circulating blood is the fastest and most effective way of viral dissemination to different tissues, including the central nervous system (CNS). (**B**) Entry into the delicate CNS parenchyma is restricted by an elaborate barrier network, most notably by the blood–brain barrier (BBB). Certain viruses can pass the BBB by the transcellular route by transferring through the lining of microvascular endothelial cells, where they may or may not replicate. In addition, viruses can pass the BBB by paracellular traversal between adjacent endothelial cells, with or without disruption of tight junctions. Viruses can also penetrate the BBB by transmigration within infected haematopoietic cells, known as the “Trojan horse” mechanism.

**Figure 2 cells-12-02307-f002:**
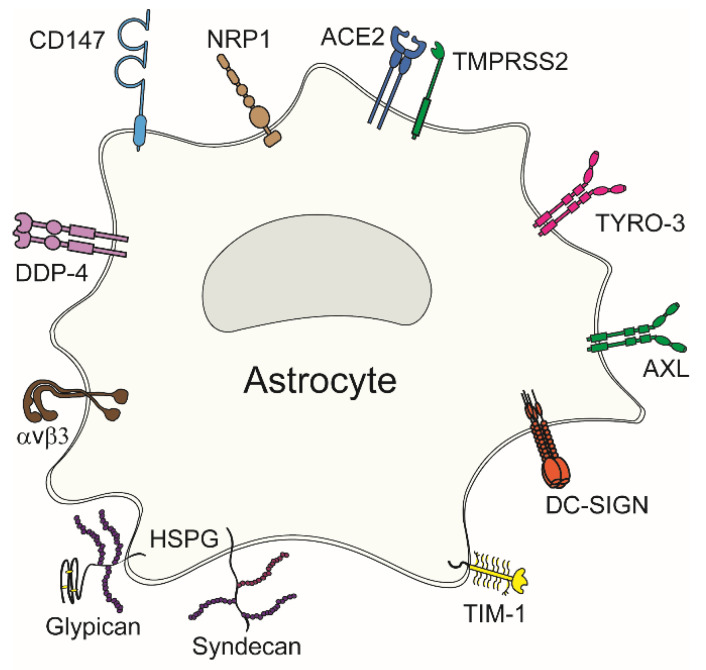
Viral entry receptors in astrocytes. Diverse plasma membrane receptors can mediate virus attachment to and entry into astrocytes. These include neuropilin-1 (NRP1), angiotensin-converting enzyme 2 (ACE2), type II transmembrane serine protease (TMPRSS2), TAM family receptors (TYRO-3 and AXL), dendritic cell (DC)-specific intercellular adhesion molecule-3-grabbing nonintegrin (DC-SIGN), T-cell Ig and mucin domain 1 (TIM-1), heparan sulphate proteoglycans (HSPGs) comprising syndecans and glypicans, αvβ3 integrin, dipeptidylpeptidase 4 (DPP4), and cluster of differentiation 147 (CD147).

**Figure 3 cells-12-02307-f003:**
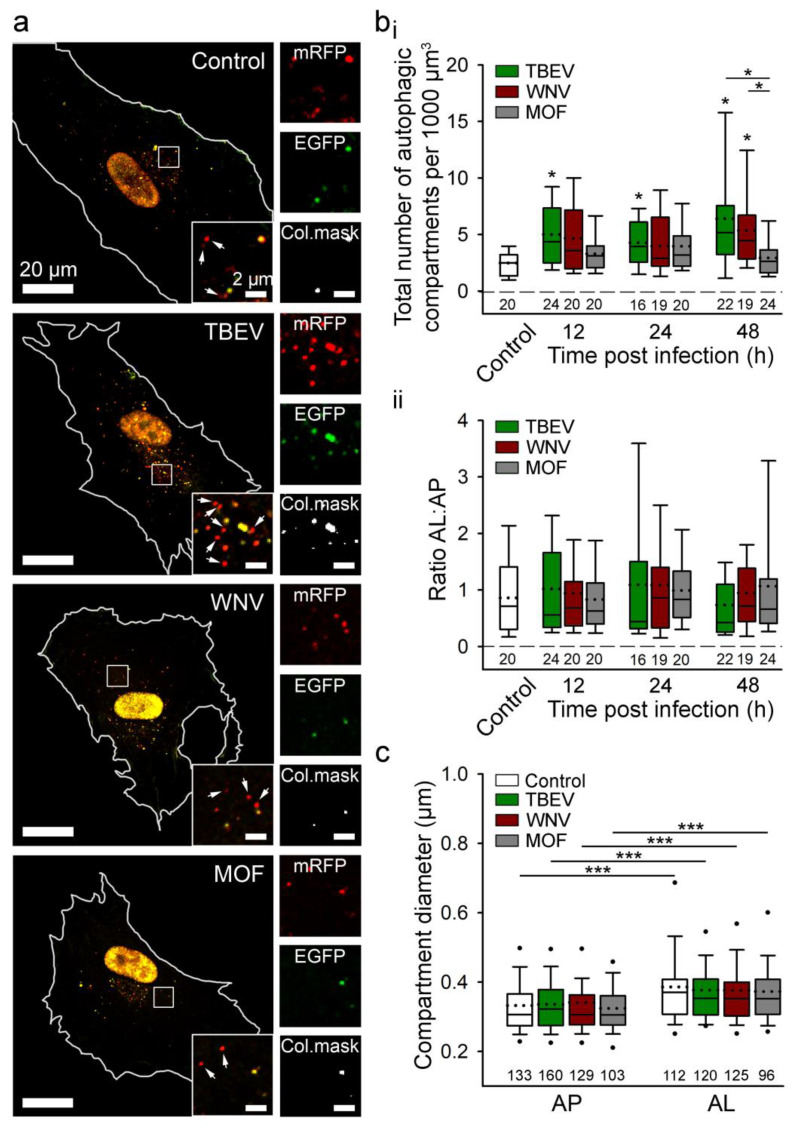
Tick-borne encephalitis virus (TBEV) and West Nile virus (WNV), but not mosquito-only flavivirus (MOF), increase autophagy in human astrocytes. (**a**) Representative fluorescence micrographs documenting mock-infected human astrocytes (Control) and astrocytes after exposure to selected flaviviruses (TBEV, WNV, and MOF) for 48 h. Cells were expressing mRFP-EGFP-LC3, where LC3, a marker of autophagic compartments, is tandem fluorescent-tagged with mRFP (red fluorescence) and EGFP (green fluorescence). The pH of autophagosomes (AP) is close to neutral, which facilitates the fluorescence of both fluorophores, resulting in yellow objects. Fusion of AP with lysosomes yields autolysosomes (AL), i.e., organelles with acidic pH, where the EGFP fluorescence is quenched, resulting in red-only objects. Selected rectangular areas within the cells, enlarged at the corners (bottom, right), show superimposed images of mRFP and EGFP fluorescence. Arrows indicate AL. Adjacent, smaller panels display mRFP and EGFP fluorescence and co-localization masks (Col.mask) (co-localized objects correspond to AP) of the enlarged images. The white outlines in the large panels show the cell shape. (**b**) The total number of autophagic compartments (**i**) and the ratio of AL to AP (which is a measure of autophagic degradation activity) (**ii**) in mock-infected cells and after infection with TBEV, WNV, and MOF at 12, 24, and 48 h post-infection (hpi). Infection with TBEV increases the total number of autophagic compartments at all three time points tested, compared with mock-infected cells (i.e., controls at 48 hpi, which were confirmed to be comparable to controls at 12 and 24 hpi) (* *p* < 0.05, one-way ANOVA followed by Dunn’s test), but does not affect the AL:AP ratio (*p*  >  0.05, one-way ANOVA). WNV infection induces an increase in the total number of autophagic compartments at 48 hpi (* *p* <  0.05, one-way ANOVA followed by Dunn’s test) and does not affect the ratio AL:AP (*p* > 0.05, one-way ANOVA). MOF infection does not affect the number of autophagic compartments or the AL: AP ratio compared with mock-infected cells at any time point tested (*p* >  0.05, one-way ANOVA). (**c**) Diameter of AP and AL in mock-infected cells and 48 hpi with selected flaviviruses. Infection with TBEV, WNV and MOF does not affect the size of the autophagic compartment compared with mock-infected cells (*p*  >  0.05, one-way ANOVA). ALs are larger than APs in all experimental conditions (*** *p* < 0.001, Mann-Whitney U test). Full lines in the boxplots represent median values, and the dotted lines correspond to average values. The numbers below the boxplots are the number of cells (**b**) or compartments (**c**) analysed for each condition. Cells were infected with TBEV and MOF at an MOI of 0.1 and with WNV at an MOI of 1. Figure and figure legend reproduced from Tavcar Verdev et al. [46] with permission, licensed under a Creative Commons Attribution 4.0 International License (http://creativecommons.org/licenses/by/4.0 (accessed on 1 August 2023)).

**Table 1 cells-12-02307-t001:** Confirmed entry receptors of neurotropic viruses in astrocytes.

Entry Receptor	Virus	References
HSPGs, αvβ3	HSV-1	[50]
ACE2	SARS-CoV-2	[36]
NRP1	SARS-CoV-2	[51]
TMPRSS2	SARS-CoV-2	[52]
AXL	ZIKV	[53]
α2β1	EV1	[54]
DC-SIGN	HIV-1	[55]

HSPGs, heparan sulphate proteoglycans; ACE2, angiotensin-converting enzyme 2; NRP1, neuropilin-1; TMPRSS2, type II transmembrane serine protease; AXL, a cell surface receptor tyrosine kinase; αvβ3, α2β1, class-I transmembrane proteins integrins; DC-SIGN, dendritic cell-specific intercellular adhesion molecule-3-grabbing nonintegrin; HSV-1, herpes simplex virus type-1; SARS-CoV-2, severe acute respiratory syndrome coronavirus 2; ZIKV, Zika virus; EV1, echovirus-1; HIV-1, human immunodeficiency virus type 1.

## Data Availability

No new data were created or analyzed in this study. Data sharing is not applicable to this article.

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
