# Peer review of "Astrocytes Are a Key Target for Neurotropic Viral Infection"

_cells, 2023, doi:10.3390/cells12182307_

Round 1
Reviewer 1 Report
This revision aims to summarize the role of astrocytes in neurotropic viral infection. The topic is very interesting; however, some aspects need to be clarified or described in more details.
1. The items under consideration are not specified based on which criteria were selected or how the information in the articles under consideration was evaluated. The lack of a methodology does not allow for verification of the authors' correct conclusions or if the information provided represents a clear search for current state of art.
2. The introduction does not reflect the purpose of this study or what it intends to add to the research niche.
3. Some sections could benefit or be more attractive by including tables summarizing the main neutropenic viruses and, for example, the routes of entry into the CNS or the mechanism of entry into astrocytes from the described receptors.
4. The relationship between astroglial aerobic glycolysis, L-lactate production by the Warburg effect and virus production briefly mentioned in section 3 is an interesting topic to describe in greater depth in a separate section.
5. Autophagy in astrocytes and its relationship with virus production could also be described in greater depth in a separate section. This topic is really interesting and would add significant value to the revision.
6. Conclusion does not reflect the whole paper and future directions or perspective are week and need a more specific projection.
Author Response
Reviewer 1:
This revision aims to summarize the role of astrocytes in neurotropic viral infection. The topic is very interesting; however, some aspects need to be clarified or described in more details.
Authors: We thank the Reviewer for his kind statement. We have followed his/her suggestions to improve the manuscript.
- Reviewer’s comment: The items under consideration are not specified based on which criteria were selected or how the information in the articles under consideration was evaluated. The lack of a methodology does not allow for verification of the authors' correct conclusions or if the information provided represents a clear search for current state of art.
Authors’ reply:
With all due respect, we would like to point out that the submitted paper is a review paper, which typically never includes the section “methodology” (Materials and Methods). We consider that our expertise and competences in the field of emerging infections of the central nervous system, especially of astrocytes, are mirrored in the published primary papers. However, the aim in this review was to highlight the most relevant information to describe astrocytes as the key target of neurotropic viral infection.
- Reviewer’s comment: The introduction does not reflect the purpose of this study or what it intends to add to the research niche.
Authors’ reply:
As mentioned in the answer to the previous comment, this is a review paper, and the introduction clearly states the scope of our manuscript is to summarize the current knowledge of the viral infections and their long-term consequences of astrocytes. These infections are still frequently overlooked, under-reported and under-diagnosed and thus represent a significant burden to human health worldwide. To avoid the repetitions with subsequent paragraphs, the key factors related to astrocytes, which can substantially contribute to the course of neuroinfection, are reviewed in the separate chapters. Moreover, to comply with the reviewer comment, we modified the abstract as follows:
Abstract: In this review we are considering astrocytes are increasingly recognized as important viral host cells in the central nervous system. These cells can, with the ability to produce relatively high quantities of new virions. In part, this can be attributed to the characteristics of the astrocyte metabolism and to the abundant and dynamic cytoskeleton network. Astrocytes are anatomically localized adjacent to interfaces between blood capillaries and brain parenchyma, and between blood capillaries and brain ventricles. Moreover, astrocytes exhibit a larger membrane interface with the extracellular space in comparison with neurones. These properties, together with the expression of various and numerous viral entry receptors, a relatively high turnover rate of endocytosis, and morphological plasticity of intracellular organelles, constitute assets that render astrocytes as important target cells in neurotropic infections. In this review, we describe factors that mediate the high susceptibility of astrocytes to viral infection and replication, including the anatomic localization of astrocytes, morphology, expression of viral entry receptors, and various forms of autophagy
- Reviewer’s comment: Some sections could benefit or be more attractive by including tables summarizing the main neutropenic viruses and, for example, the routes of entry into the CNS or the mechanism of entry into astrocytes from the described receptors.
Authors’ reply:
We are grateful for the reviewer’s suggestion. The main neurotropic viruses that infect astrocytes have been described in our recent paper entitled “Immune Functions of Astrocytes in Viral Neuroinfections” (Int. J. Mol. Sci. 2023, 24, 3514). We refer to this paper on page 4 “Numerus neurotropic viruses may infect astrocytes, and consequently indirectly alter cellular functions of neighbouring neurons and the BBB-forming endothelial cells ([50] and references within)”. Hence, in our view it would be redundant to repeat this information. In addition, the entry routes are also mentioned in the afore-mentioned paper, with relevant references.
Moreover, we would like to direct the attention of the reviewer to the text on page 7 that reads: “The rate-determining steps include attachment of viruses to the host cell receptors and the ensuing endocytosis or virus entry into the host cell by directly penetrating the plasma membrane, replication, and assembly of new virions and subsequently on their egress.”
However, to comply with the reviewer’s suggestion, we now briefly describe the complete life cycle of an endocytosed virus on page 8: “The viral replication cycle comprise of four major steps: attachment and entry into a target cell, replication of the viral genome, maturation of viral proteins and genome packaging into infectious progeny, egress to the extracellular space and dissemination to the next target cell. In the extracellular environment host viruses are routinely confronted with antibody neutralization, pattern recognition receptors and the downregulation of the cell surface receptors required for their entry.”
In addition, we have included the table summarizing confirmed virus entry receptors in astrocytes (page 4):
Table 1. Confirmed entry receptors of neurotropic viruses in astrocytes.
Entry receptor |
Virus |
References |
HSPGs, αvβ3 |
HSV-1 |
[57] |
ACE2 |
SARS-CoV-2 |
[36] |
NRP1 |
SARS-CoV-2 |
[68] |
TMPRSS2 |
SARS-CoV-2 |
[63] |
AXL |
ZIKV |
[87] |
α2β1 |
EV1 |
[89] |
DC-SIGN |
HIV-1 |
[100] |
HSPGs, heparan sulphate proteoglycans; ACE2, angiotensin converting enzyme 2; NRP1, neuropilin-1; TMPRSS2, type II transmembrane serine protease; AXL, a cell surface receptor tyrosine kinase; αvβ3, α2β1, αvβ6, α5β1, α3β1, β1, class-I transmembrane proteins integrins; DC-SIGN, dendritic cell-specific intercellular adhesion molecule-3-grabbing nonintegrin; HSV-1, herpes simplex virus type-1; SARS-CoV-2, severe acute respiratory syndrome coronavirus 2; ZIKV, Zika virus; EV-1, echovirus-1; HIV-1, human immunodeficiency virus type 1
- Reviewer’s comment: The relationship between astroglial aerobic glycolysis, L-lactate production by the Warburg effect and virus production briefly mentioned in section 3 is an interesting topic to describe in greater depth in a separate section.
Authors’ reply:
We agree with the reviewer that description of this relationship would be an interesting topic. However, the relation between the astrocyte metabolism and viral infections is an emerging topic and one would need to consider aerobic glycolysis to a greater detail, by discussing various ramifications of the glycolytic flux, relevant for nucleic acid synthesis, the coupling with TCA. This would generate a new focus of the paper which is beyond the main message of the manuscript Moreover, we have already covered some of the issues in previous papers, meaning that this will generate some repetition.
- Reviewer’s comment: Autophagy in astrocytes and its relationship with virus production could also be described in greater depth in a separate section. This topic is interesting and would add significant value to the revision.
Authors’ reply: Autophagy is now described more extensively in chapter 6 (pages 8 to 11) that is clearly separated from other chapters. In particular, the legend of Figure 3, which is dedicated to show the modulation of autophagy in astrocytes by some viruses (e.g. tick-born encephalitis virus and West Nile virus), but not all of them (i.e. mosquito only flavivirus), has been revised and now reads:
Figure 3. Tick-borne encephalitis virus (TBEV), West Nile virus (WNV), but not mosquito-only flavivirus (MOF), increase autophagy in human astrocytes. (a) Representative fluorescence micrographs documenting mock-infected human astrocytes (Control) and astrocytes after exposure to selected flaviviruses (TBEV, WNV, and MOF) for 48 h. Cells were expressing mRFP-EGFP-LC3, where LC3, a marker of autophagic compartments, is tandem fluorescent-tagged with mRFP (red fluorescence) and EGFP (green fluorescence). The pH of autophagosomes (AP) is close to neutral, which facilitates the fluorescence of both fluorophores, resulting in yellow objects. Fusion of AP with lysosomes yields autolysosomes (AL), i.e., organelles with acidic pH, where the EGFP fluorescence is quenched, resulting in red-only objects. Selected rectangular areas within the cells, enlarged at the corners (bottom, right), show superimposed images of mRFP and EGFP fluorescence. Arrows indicate AL. Adjacent, smaller panels display mRFP and EGFP fluorescence as well as co-localization masks (Col.mask) (co-localized objects correspond to AP) of the enlarged images. The white outlines in the large panels show the cell shape. (b) The total number of autophagic compartments (i) and the ratio of AL to AP (which is a measure of autophagic degradation activity) (ii) in mock-infected cells and after infection with TBEV, WNV, and MOF at 12, 24, and 48 h post infection (hpi). Infection with TBEV increases the total number of autophagic compartments at all three time points tested, compared with mock-infected cells (i.e. controls at 48 hpi, which were confirmed to be comparable to controls at 12 and 24 hpi) (*P < 0.05, one-way ANOVA followed by Dunn's test), but does not affect the AL:AP ratio (P > 0.05, one-way ANOVA). WNV infection induces an increase in the total number of autophagic compartments at 48 hpi (*P < 0.05, one-way ANOVA followed by Dunn’s test), and does not affect the ratio AL:AP (P > 0.05, one-way ANOVA). MOF infection does not affect the number of autophagic compartments or the AL:AP ratio compared with mock-infected cells at any time point tested (P > 0.05, one-way ANOVA). (c) Diameter of AP and AL in mock-infected cells and 48 hpi with selected flaviviruses. Infection with TBEV, WNV and MOF does not affect the size of the autophagic compartment compared with mock-infected cells (P > 0.05, one-way ANOVA). ALs are larger than APs in all experimental conditions (***P < 0.001, Mann-Whitney U test). Full lines in the boxplots represent median values and the dotted lines correspond to average values. The numbers below the boxplots are the number of cells (b) or compartments (c) analysed for each condition. Cells were infected with TBEV and MOF at an MOI of 0.1 and with WNV at an MOI of 1 Figure and figure legend reproduced from Tavcar Verdev et al. [46] with permission, licensed under a Creative Commons Attribution 4.0 International License (http://creativecommons.org/licenses/by/4.0).
- Reviewer’s comment: Conclusion does not reflect the whole paper and future directions or perspective are week and need a more specific projection.
Authors’ reply: Our view is that the conclusion section should be brief. In addition, we would like to point out that as of now, no effective antiviral drugs have been developed for most viral infections, nor are we in a possession of astrocyte-specific therapies or drugs. To comply with the reviewer’s suggestion and at the same time avoid speculation, we have decided to end the section Conclusion with the following text (page 11):
Of note, understanding the mechanisms of virus entry and replication in astrocytes may lead to therapeutic strategies that would reduce their capacity as a CNS viral reservoir, as well as help with virus-associated astrocyte demise. This would be a major step forward, as currently, no effective antiviral drugs are available for many viral infections, nor are we in a possession of astrocyte-specific therapies or drugs.
Reviewer 2 Report
Authors of the review entitled „Astrocytes Are a Key Target for Neurotropic Viral Infection” choose a very important topic for their work. It should be underlined that espetially topic of manuscript is very significant and smartly chosen. The work is well organized and written although some issues require improvement.
Major:
1. Although figures are very well prepared in terms of their content and “artistic value” their resolution must be improved. When zoomed just a little bit they are pixelized.
2. One general problem of work is the fact that multiple times authors use the general term without specification (tight junction, cytoskeleton remodeling proteins, hematopoietic cells etc.). It should be improved based on already cited literature but also on new references.
3. It will be valuable to address methods applied in studies of pathogen-receptor interactions. Including single-cell force spectroscopy (10.1371/journal.pone.0086219), optical tweezers (ie. 10.1371/journal.pone.0086219) and other.
4. Adhesion of viruses is a physical process. Consequently authors address cytoskeleton in the context of viruses adhesion. Since those both processes are directly connected, it might be valuable to consider the participation physical factors already investigated in brain (10.1007/s43440-021-00315-2) in the process of viral infection (i.e. sars-cov2 10.1038/s41422-021-00558-x). It will be very valuable to write a little bit more about role of physical interactions, adhesion proteins and cytoskeleton. Those factors are recently investigated with increasing intensity. Addressing them will be for sure the advantage for this paper.
5. Although figure 2 is quiet nice to organize knowledge and memorize receptors engaged in this process it could be valuable to provide table summarizing information about receptors responsible for viruses entry to astrocytes. Treat it not as an obligatory point but something what should make yours review reader-friendlier.
6. One point which is really missing is chapter about potential therapeutic strategies against CNS viral infections targeting viral interactions with astrocytes.
Minor:
1. Page 1 24-25 “most of the 25 virus families implicated in human disease 24 include representatives that have been associated with CNS disorders in humans [3,4]” – this sentence is not clear. Does it state that majority of human viruses are associated with CNS disorders?
2. Page 1 26 “ infection of the CNS infection” – correct it
3. Page 1 “(with the exception of extrinsic 38 viral contamination of the CNS during neurosurgical procedures).” Is neurosurgical procedure the only exception?
4. Page 2 authors state “Factors that restrict an infection from spreading 45 beyond an epithelial surface have been reviewed extensively [7].” Word extensively suggests something prominent but authors cite only one work. Please provide more references or make this statement milder.
5. Page 3 Please specify the proteins forming tight-junction in BBB.
6. Page 3 Do authors think that term express TJ is proper? Wouldn’t it be better to said express TJs proteins?
7. Page 3 Again, precise which cytoskeleton-regulating proteins in BBB
8. Page 3 “The Trojan horse mechanism of viral infection of the CNS refers to the haematopoietic cells infected with virus 112 that transport virus into the CNS [23,25,31,32].” Authors cite 4 works but do not precise which exact hematopoietic cells play role in this important process.
9. Page 3 “After crossing the BBB, viruses may use multiple routes to invade the CNS [33]” - this sentence do not provide any important information into work. Please precise its message or delete it.
10. Page 3,4 “Moreover, compared with neurons, astrocytes have a much higher surface area-to-volume ratio” this information is provided twice. Please left it only in one point of manuscript.
11. Page 4 “and during infections contains infectious viral particles.” Provide reference to this statement
12. Page 4 Please provide references to those statement “Viral infection of astrocytes affects a number of cellular processes, altering their morphologic and physiologic properties, including rearrangement and reorganization of the cytoskeleton, modulation of vesicular traffic with endocytosis and cellular organelles, affecting the transcription of genes with an altered expression profile of cytokines, affecting autophagy and resulting in damage to cellular DNA.” Just examples doi.org/10.1007%2Fs00705-021-05025-x, 10.1371/journal.pone.0086219 but here are far more another papers in this field
13. Page 6 “e SARS- 228 Cov2” please correct Cov2
14. Page 7 Please precise with which paritculare integrins e Picornaviridae and Herpesviridae families members interacts. Does aforementioned αvβ3, αvβ5 and αvβ8 belong to them?
Author Response
Reviewer 2:
Major:
- Reviewer’s comment:Although figures are very well prepared in terms of their content and “artistic value” their resolution must be improved. When zoomed just a little bit they are pixelized.
Authors’ reply: We are grateful to the reviewer for the comment, but would like to mention that this is likely due to the down-sampling intended for a more compacted file preparation for the reviewing process. We have submitted a 600 dpi images, which is sufficient for printing high quality graphics as well as smaller font sizes.
- Reviewer’s comment:One general problem of work is the fact that multiple times authors use the general term without specification (tight junction, cytoskeleton remodeling proteins, hematopoietic cells etc.). It should be improved based on already cited literature but also on new references.
Authors’ reply: In line with this suggestion, in the revised manuscript we now specify both tight junction and hematopoietic cells, when they are first mentioned in the text. Of note, we do not use the term “cytoskeleton remodeling proteins” in the manuscript.
We have included the following text on page 3:
“The structural components that form the BBB are endothelial cells, which, unlike other vascular endothelial cells, express tight junction (TJ) proteins that seal the paracellular space between adjoining endothelial cells. These proteins include claudins, occludin, tricelllin, lipolysis-stimulated lipoprotein receptor, junctional adhesion molecules and zonula occludens proteins; the latter form a structural link to the actin cytoskeleton and actin binding proteins. In addition to endothelial cells, astrocytes and pericytes are critical for the development and maintenance of the BBB by regulating the TJs [26–28] (see also Figure 1B).”
and we have modified the following sentence related to leukocytes as the most common (but not exclusive) haematopoietic cell type involved in the Trojan horse mechanism of viral infection of the CNS (page 3):
“The Trojan horse mechanism of viral infection of the CNS refers to the haematopoietic cells (e.g., leukocytes; Figure 1) infected with virus that transport virus into the CNS [23,25,31,32].
- Reviewer’s comment:It will be valuable to address methods applied in studies of pathogen-receptor interactions. Including single-cell force spectroscopy (10.1371/journal.pone.0086219), optical tweezers (ie. 10.1371/journal.pone.0086219) and other.
Authors’ reply: With all due respect, we feel that, due to the wide range of various methods, that are used to study pathogen-receptor interactions, this topic would be a relevant self-standing review within a special issue/journal focused on methodological approaches to study the viral infections of cells.
- Reviewer’s comment: Although figure 2 is quiet nice to organize knowledge and memorize receptors engaged in this process it could be valuable to provide table summarizing information about receptors responsible for viruses entry to astrocytes. Treat it not as an obligatory point but something what should make yours review reader-friendlier.
Authors’ reply: In line with the suggestion, we have included the table summarizing confirmed virus entry receptors in astrocytes (page 4):
Table 1. Confirmed entry receptors of neurotropic viruses in astrocytes.
Entry receptor |
Virus |
References |
HSPGs, αvβ3 |
HSV-1 |
[57] |
ACE2 |
SARS-CoV-2 |
[36] |
NRP1 |
SARS-CoV-2 |
[68] |
TMPRSS2 |
SARS-CoV-2 |
[63] |
AXL |
ZIKV |
[87] |
α2β1 |
EV1 |
[89] |
DC-SIGN |
HIV-1 |
[100] |
HSPGs, heparan sulphate proteoglycans; ACE2, angiotensin converting enzyme 2; NRP1, neuropilin-1; TMPRSS2, type II transmembrane serine protease; AXL, a cell surface receptor tyrosine kinase; αvβ3, α2β1, αvβ6, α5β1, α3β1, β1, class-I transmembrane proteins integrins; DC-SIGN, dendritic cell-specific intercellular adhesion molecule-3-grabbing nonintegrin; HSV-1, herpes simplex virus type-1; SARS-CoV-2, severe acute respiratory syndrome coronavirus 2; ZIKV, Zika virus; EV-1, echovirus-1; HIV-1, human immunodeficiency virus type 1
- Reviewer’s comment: One point which is really missing is chapter about potential therapeutic strategies against CNS viral infections targeting viral interactions with astrocytes.
Authors’ reply: We agree with the proposal of the reviewer, however, we would like to point out that as of now, no effective antiviral drugs have been developed for most viral infections, nor are we in a possession of astrocyte-specific therapies or drugs. Therefore, to avoid speculation, we have decided to end the section Conclusion with the following text (page 11):
Of note, understanding the mechanisms of virus entry and replication in astrocytes may lead to therapeutic strategies that would reduce their capacity as a viral reservoir, as well as help with virus-associated astrocyte loss. This would be a major step forward, as currently, no effective antiviral drugs are available for many viral infections, nor are we in a possession of astrocyte-specific therapies or drugs.
Minor:
- Reviewer’s comment:Page 1 24-25 “most of the 25 virus families implicated in human disease include representatives that have been associated with CNS disorders in humans [3,4]” – this sentence is not clear. Does it state that majority of human viruses are associated with CNS disorders?
Authors’ reply: No, the statement should state that most of 25 virus families implicated in human disease have at least some representatives with the potential to cause CNS disorders in humans. Hence, to clarify this statement we have altered the sentence on page 1 that now reads:
“Infectious diseases of the central nervous system (CNS) are most commonly caused by various types of viruses [1,2]; most of the 25 virus families implicated in human disease include certain representatives that have been associated with CNS disorders in humans [3,4].”
- Reviewer’s comment:Page 1 26 “ infection of the CNS infection” – correct it
Authors’ reply: Corrected.
- Reviewer’s comment:Page 1 “(with the exception of extrinsic viral contamination of the CNS during neurosurgical procedures).” Is neurosurgical procedure the only exception?
Authors’ reply: The sentence on page 2 “The CNS is typically not the site of initial virus entry, with the exception of extrinsic viral contamination of the CNS during neurosurgical procedures).”
is now modified to:
”The CNS is typically not the site of initial virus entry, with the exception of extrinsic viral contamination of the CNS during head trauma, neurosurgical procedures, via medical implants, and similar.”
- Reviewer’s comment Page 2 authors state “Factors that restrict an infection from spreading 45 beyond an epithelial surface have been reviewed extensively [7].” Word extensively suggests something prominent but authors cite only one work. Please provide more references or make this statement milder.
Authors’ reply: The sentence “Factors that restrict an infection from spreading beyond an epithelial surface have been reviewed extensively [7].”
Is now modified and reads:
“Factors that restrict an infection from spreading beyond an epithelial surface have been described in detail previously [7].”
- Reviewer’s comment:Page 3 Please specify the proteins forming tight-junction in BBB.
Authors’ reply: In line with this suggestion we modified the text on page 3, which now reads:
“The structural components that form the BBB are endothelial cells, which, unlike other vascular endothelial cells, express tight junction (TJ) proteins that seal the paracellular space between adjoining endothelial cells. These proteins include claudins, occludin, tricelllin, lipolysis-stimulated lipoprotein receptor, junctional adhesion molecules and zonula occludens proteins; the latter form a structural link to the actin cytoskeleton and actin binding proteins. In addition to endothelial cells, astrocytes and pericytes are critical for the development and maintenance of the BBB by regulating the TJs [26–28] (see also Figure 1B).”
- Reviewer’s comment: Page 3 Do authors think that term express TJ is proper? Wouldn’t it be better to said express TJs proteins?
Authors’ reply: Corrected.
- Reviewer’s commentPage 3 Again, precise which cytoskeleton-regulating proteins in BBB
Authors’ reply: Corrected.
- Reviewer’s commentPage 3 “The Trojan horse mechanism of viral infection of the CNS refers to the haematopoietic cells infected with virus that transport virus into the CNS [23,25,31,32].” Authors cite 4 works but do not precise which exact hematopoietic cells play role in this important process.
Authors’ reply: We have modified the following sentence related to leukocytes as the most common (but not exclusive) haematopoietic cell type involved in the Trojan horse mechanism of viral infection of the CNS (page 3):
“The Trojan horse mechanism of viral infection of the CNS refers to the haematopoietic cells (e.g., leukocytes; Figure 1) infected with virus that transport virus into the CNS [23,25,31,32].
- Reviewer’s comment:Page 3 “After crossing the BBB, viruses may use multiple routes to invade the CNS [33]” - this sentence do not provide any important information into work. Please precise its message or delete it.
Authors’ reply: In line with the Reviewer’s suggestion, we have modified the sentence that now reads (page 3):
After crossing the BBB, viruses may disperse through the CNS [33].
- Reviewer’s comment:Page 3,4 “Moreover, compared with neurons, astrocytes have a much higher surface area-to-volume ratio” this information is provided twice. Please left it only in one point of manuscript.
Authors’ reply: In line with this suggestion, we have modified the sentence on page 4 from:
“The basic morphological backbone of astrocytes has been shown to be similar to that of neurons, yet detailed morphology reveals cell expansions that may branch to thousands of smaller processes [42]. Moreover, compared with neurons, astrocytes have a much higher surface area-to-volume ratio [42] and therefore have a much larger membrane interface with the extracellular space, which is estimated to occupy ~20% of brain tissue [49], and during infections contains infectious viral particles.”
To:
from “The basic morphological backbone of astrocytes has been shown to be similar to that of neurons, yet detailed morphology reveals cell expansions that may branch to thousands of smaller processes [42], enabling astrocytes to have a much larger membrane interface with the extracellular space. Extracellular space, which is estimated to occupy ~20% of brain tissue [49], during infections contains complete viral particles and extracellular vesicles containing infective viral genomes and quasi-enveloped viruses [45].”
- Reviewer’s comment: Page 4 “and during infections contains infectious viral particles.” Provide reference to this statement
Authors’ reply: Done, please see the answer to the previous comment.
- Reviewer’s comment:Page 4 Please provide references to those statement “Viral infection of astrocytes affects a number of cellular processes, altering their morphologic and physiologic properties, including rearrangement and reorganization of the cytoskeleton, modulation of vesicular traffic with endocytosis and cellular organelles, affecting the transcription of genes with an altered expression profile of cytokines, affecting autophagy and resulting in damage to cellular DNA.” Just examples doi.org/10.1007%2Fs00705-021-05025-x, 10.1371/journal.pone.0086219 but here are far more another papers in this field
Authors’ reply: We have included suitable references, as suggested.
- Reviewer’s comment:Page 6 “e SARS-Cov2” please correct Cov2
Authors’ reply: Corrected.
- Reviewer’s comment:Page 7 Please precise with which paritculare integrins e Picornaviridae and Herpesviridae families members interacts. Does aforementioned αvβ3, αvβ5 and αvβ8 belong to them?
Authors’ reply: Picornaviruses and herpesviruses interact with several integrins, hence the sentence on page 7 “Viruses from the Picornaviridae and Herpesviridae families, some of which were shown to infect astrocytes, interact with several members of the integrin family [89].”
was modified:
“Viruses from the Picornaviridae and Herpesviridae families, some of which were shown to infect astrocytes, interact with several members of the integrin family, including αvβ3, αvβ6, α2β1, α5β1 in the case of picornaviruses and β1, αvβ3, α3β1, α2β1 in the case of herpesviruses [89].”
Reviewer 3 Report
Cells-2271155
"Astrocytes Are a Key Target for Neurotropic Viral Infection"
Authors: Maja Potokar, Robert Zorec and Jernej Jorgačevski
This is a very interesting manuscript on the contribution of astrocytes to the entry and as potential target of viral replication and infection of CNS. The manuscript is generally well written but some important questions that, in my opinion would greatly improve the manuscript, have not been addressed.
Major comments and suggestions:
1- In the subtitle “Viral Attachment to Entry Receptors in Astrocytes” the authors present a figure with an astrocyte and all the potential surface proteins that can be target by a virus once it enters the CNS, but not all the receptors were referend with examples as targets for the virus to enter astrocytes. Only some were referred.
2- On the subtitle “Endocytosis and Remodeling of Intracellular Organelle Traffic in Infected Astrocytes” several cellular pathways, that can be used by the virus to survive, and eventually replicate, are described. However, there is not a single example of a “complete life cycle of a virus” in the astrocytes. This should be described to support all the work and would greatly improve the manuscript.
Best regards.
Author Response
Reviewer 3:
"Astrocytes Are a Key Target for Neurotropic Viral Infection"
Authors: Maja Potokar, Robert Zorec and Jernej Jorgačevski
This is a very interesting manuscript on the contribution of astrocytes to the entry and as potential target of viral replication and infection of CNS. The manuscript is generally well written but some important questions that, in my opinion would greatly improve the manuscript, have not been addressed.
Authors: We thank the Reviewer for his kind statement. We have followed his/her suggestions to clarify certain aspects.
Major comments and suggestions:
- Reviewer’s comment: In the subtitle “Viral Attachment to Entry Receptors in Astrocytes” the authors present a figure with an astrocyte and all the potential surface proteins that can be target by a virus once it enters the CNS, but not all the receptors were referend with examples as targets for the virus to enter astrocytes. Only some were referred.
Authors’ reply: We have carefully read this chapter and in line with the reviewer’s comment, we included missing references.
- Reviewer’s comment: On the subtitle “Endocytosis and Remodeling of Intracellular Organelle Traffic in Infected Astrocytes” several cellular pathways, that can be used by the virus to survive, and eventually replicate, are described. However, there is not a single example of a “complete life cycle of a virus” in the astrocytes. This should be described to support all the work and would greatly improve the manuscript.
Authors’ reply: We cannot completely agree with the reviewer that the virus cycle was not mentioned even once, since the text on page 7 reads as “The rate-determining steps include attachment of viruses to the host cell receptors and the ensuing endocytosis or virus entry into the host cell by directly penetrating the plasma membrane, replication, and assembly of new virions and subsequently on their egress.”
In line with the reviewer’s comment, we now extended this text to describe the complete life cycle of an endocytosed virus on page 8:
“The viral replication cycle comprises four major steps: attachment and entry into a target cell, replication of the viral genome, maturation of viral proteins and genome packaging into infectious progeny, egress to the extracellular space and dissemination to the next target cell. In the extracellular environment of a host viruses are routinely confronted with antibody neutralization, pattern recognition receptors and the downregulation of the cell surface receptors required for entry.”
Round 2
Reviewer 1 Report
Authors have take into consideration my suggestions. I have not more comments. I believe that the article is suitable to be published.
Author Response
Thank you for your kind response.
Reviewer 3 Report
The manuscript was significantly improved and the questions raised were answered.
Author Response
Thank you for your kind response.